# Situating Speech Synthesis: Investigating Contextual Factors in the Evaluation of Conversational TTS

*Harm Lameris, Ambika Kirkland, Joakim Gustafson, Éva Székely*

Division of Speech, Music and Hearing, KTH Royal Institute of Technology, Stockholm, Sweden

{lameris, kirkland, jkgu, szekely}@kth.se

## Abstract

Speech synthesis evaluation methods have lagged behind the development of TTS systems, with single sentence read-speech MOS naturalness evaluation on crowdsourcing platforms being the industry standard. For TTS to successfully be applied in social contexts, evaluation methods need to be socially embedded in the situation where they will be deployed. Due to the time and cost constraints of conducting an in-person interaction evaluation for TTS, we examine the effect of introducing situational context and preceding sentence context to participants in a subjective listening experiment. We conduct a suitability evaluation for a robot game guide that explains game rules to participants using two synthesized spontaneous voices: an instruction-specific and a general spontaneous voice. Results indicate that the inclusion of context influences user ratings, highlighting the need for context-aware evaluations. However, the type of context did not significantly affect the results.

**Index Terms**: speech synthesis, text to speech, evaluation, social, context

## 1. Introduction

With the introduction of voice assistants and other devices utilizing speech synthesis has led to a dramatic expansion in the use cases and purposes of text-to-speech (TTS) systems. In recent years, with the rise of deep-learning-based architectures and the associated increase in naturalness, speech synthesis research has increasingly focused on enhancing the modeling of prosodic variation and on aspects such as synthesizing style and emotion [1, 2, 3]. However, evaluation methods for synthesized speech have not kept up with the rapid development and diverse applications of synthetic voices. Current evaluation techniques primarily rely on absolute category rating (ACR) with metrics such as Mean Opinion Score (MOS), the slightly more fine-grained MUSHRA, or the comparative MOS (CMOS) [4]. The use of these methods, which have remained largely unchanged since they were developed by the International Telecommunication Union (ITU) in the 1990s [5]. These methods have been criticized for oversimplifying complex concepts like naturalness or quality into a single measure [4] and for the frequent omission of essential information in research reports, such as the specific questions posed to participants [6]. Wagner et al. [5] argue that a more fitting evaluation for speech synthesis is rating appropriateness socially embedded in a context. It is important to ensure that the TTS voice is able to convey the kinds of communicative functions [7] you would need, given the intended end application. Dall et al. [8] found that participants' expectations of speaking style influenced their ratings of naturalness, with spontaneous human speech being rated as more natural than read human speech, even when instructed to rate the speech in a read-aloud context. Evaluations integrating some form of context are increasingly used in speech synthesis, especially focusing on the preceding speech context. In [9] the question of naturalness in single utterances, entire paragraphs, and selected stretches of preceding speech or text as context was evaluated, and significant differences between the categories were found, as well as a lack of correlation for the ratings. It was additionally found that MOS ratings increased in context both for real and synthesized speech despite using identical audio, except when real speech was used as context for a synthesized stimulus. O'Mahony [10] replicated the increase in MOS scores when presenting context in the evaluation, and additionally concluded that this was not influenced by the between-sentence context dependency of stimulus the text and that scores decreased for non-canonical prosody. In [11] lower ratings were reported for paragraphs compared to isolated sentences when participants conducted MOS evaluations of synthesized speech modified with control tags to mark focus in prosody. Despite this, the focus-aware speech achieved a higher MOS than unmodified speech. While these studies introduce context as part of TTS evaluation, they primarily consider the preceding text or audio as context and do not take into account the conceptual framing of application situations, specific needs, or listener preferences [5] which can be especially important for conversational TTS aimed at social robots. Furthermore, most evaluations utilize read-speech voices, which lack convincing prosody for longer speech segments, [12]. Although interaction evaluation designs, such as those presented in [13], would be ideal, they demand resources that may not be available during the development of speech synthesis models.

In this paper, we explore the effects of situational context and preceding sentence context on listeners' perception of synthesized speech trained on ecologically valid conversational data. The spontaneous TTS system used in the study was trained on a corpus recorded in actual interactions, instead of using scripted dialogue. This choice was motivated by previous studies which show that only some pragmatic uses of prosody could be reproduced in read interactions [14]. To determine the most suitable voice for a game guide social robot, we conduct four subjective listening experiments evaluating a spontaneous instruction-specific voice and a general spontaneous voice. We performed a naturalness MOS without context, and suitability evaluations in which we provide participants with one of the following: situational context, preceding sentence context, or full context, in which participants receive both situational and preceding sentence context. Results show that while both voices are rated as equal in terms of naturalness in the MOS evaluation, the instruction-specific voice was rated significantly better than the general voice in conditions where context was provided. However, the type of context did not make a difference.

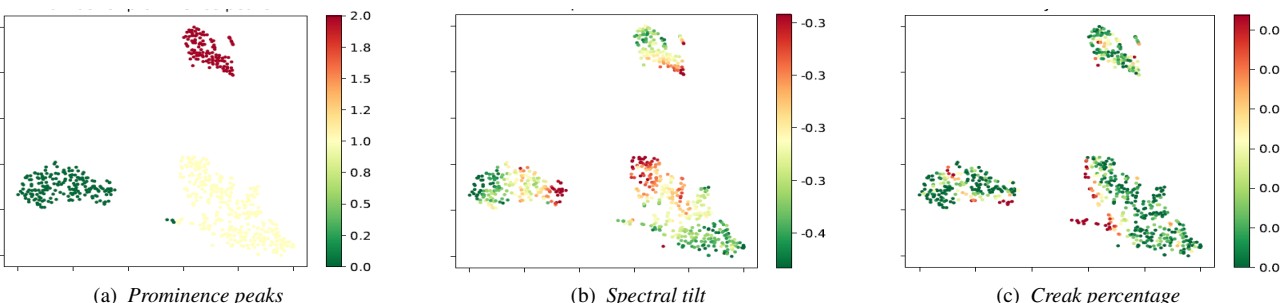

|  (a) *Prominence peaks* | (b) *Spectral tilt* | (c) *Creak percentage* |

Figure 1: *The Starmap [15] t-SNE plot for prosodic features.*

## 2. Method

### 2.1. Data

The data used in this study was taken from a multimodal corpus that consists of 15 interactions between a human moderator and two users who were tasked with the interior decoration of an apartment using a GUI on a large touch screen [16]. Participants were prompted to design a living space they would share for three months while being filmed for a hypothetical reality television program. The same moderator conducted each interaction, with data collection aimed at developing a social robot for moderating similar collaborative tasks. While the moderator followed a general outline of topics to cover in each interaction, he was not given specific instructions on what to say, allowing for spontaneous yet pre-planned extemporaneous conversations during the interactions.

The multimodal corpus contains several distinct interaction phases. In the first phase, the moderator started with *small talk* which was intended to help acclimatize the participants. In the phase, the moderator discussed various topics such as tidiness and conflicts in living situations with the participants. The second phase involved the moderator providing *instructions* on the experiment's setup, where participants collaborated on designing a living space for a hypothetical reality television series lasting three months. The third phase saw the moderator assuming the role of an interior decorator, offering design advice to participants. The final phase involved the moderator commenting on the participants' final choices and conducting a debriefing session [16].

For the TTS model, we used speech data exclusively from the male moderator who spoke General American English. The data was automatically segmented into breath groups, stretches of speech uttered between two breaths, lasting 1 to 10 seconds, and transcribed using ASR. Manual corrections were made to the transcription, with filled pauses, audible breaths, turn-internal pauses, and turn-endings represented by specific tokens. This transcription enables explicit control over the conversational agent's manner of speaking when used as input to a TTS system dialogue. Alongside spontaneous speech data, we included read-speech audio of the moderator reading 1129 sentences from the CMU Arctic dataset [17] and 1132 sentences from online news articles. This resulted in a TTS corpus of approximately 8 hours, with 2 hours and 26 minutes of read speech and 5 hours and 40 minutes of spontaneous speech.

To incorporate the preceding sentence from the recording as context, we extracted excerpts from two held-out recordings within the multimodal corpus. These sentences originated from the second phase, where the moderator informed and instructed participants about the experiment's setup. Audio excerpts were restricted to durations between 2 and 8 seconds and had to satisfy two criteria: no interruptions or overlapping speech from other participants and being immediately followed by a sentence relating to the same topic. Since the excerpts were part of a three-party interaction, single sentences were used as context, as longer stretches failed to meet these criteria.

### 2.2. Data annotation

The data was preprocessed identically to [18] which can be found in the supplemental material. We selected 490 utterances based on their length, ranging from 4-10 seconds, which were then annotated using Starmap, a human-in-the-loop annotation tool [15] that uses t-SNE visualization [19] on utterance-level prosodic features to assist in the exploration of corpora in a linguistically-informed manner. Five prosodic features which were hypothesized to contribute to the distinction of the previously described phases were calculated in the following manner: the *mean energy*, *number of prominence peaks*, and the estimated *speech rate* were extracted continuous wavelet transformation based hierarchical prosody representation [20] while the prominence peaks were extracted using high-level hierarchical scales as described in [15, 20]. Additionally, we included per-utterance *spectral tilt* over voiced segments and *creak percentage*. The spectral tilt was calculated using the Python package Parselmouth [21], while the creak percentage was determined following [22] by extracting the duration of creaky voice per file using DeepFry [23] and calculating the creak percentage as:

$$\text{creak percentage} = \frac{\text{total creak duration}}{\text{total duration}} \qquad (1)$$

We used the t-SNE dimensionality reduction method [19] as implemented in Starmap to visualize these prosodic features in a plot (Figure 1) and enable selection of utterances based on pragmatic function. Using the prompt selection tool from Starmap, we annotated 490 utterances according to their pragmatic function based on both the prosodic features and semantic content. The pragmatic functions we used for annotations were small talk (99 utterances), instructions (130 utterances), advice/guidance (163 utterances), and self-directed speech during decision making (98 utterances).

While [18] use the four pragmatic functions in their evaluation, we focus on the spontaneous *instruction* style, as this most closely corresponds to the function of a social-robot game guide.

### 2.3. Model architecture

We use the architecture and training steps of [18], which is a modified version of the Tacotron2 [24] architecture imple-

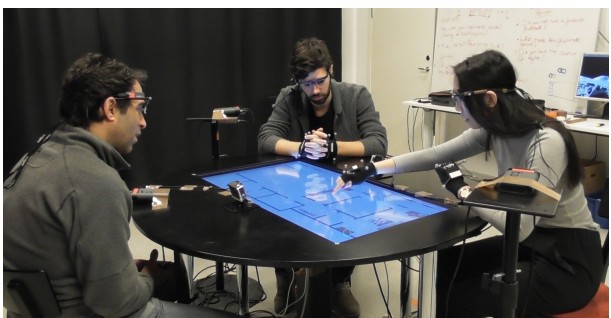

Figure 2: *Data collection photo, courtesy of the authors [16].*

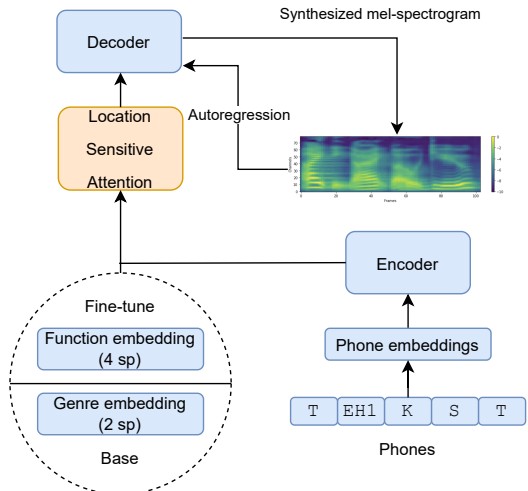

Figure 3: *The model architecture.*

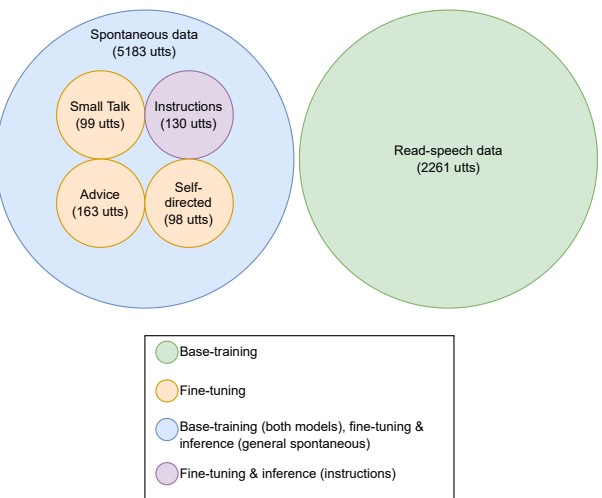

Figure 4: *An overview of the data used during training.*

mented in PyTorch[1]. The model incorporates an 8-dimensional speaker-like embedding inspired by [25] to indicate the pragmatic function (Figure 3). This embedding is concatenated with the encoded text of each utterance and passed to the attention and decoder blocks, increasing the parameter count from 28.19M to 28.26M. When base training is completed, the embeddings are reinitialized, even if the number of pragmatic functions is identical.

The base training for the model is performed on the whole corpus, to which we add two embeddings that indicate whether utterances originate from the read-speech part or the spontaneous part of the corpus. The base model is trained 70k iterations on 4 NVIDIA GeForce RTX 3090 12 GB GPUs with batch size 28 and with 5% of the data withheld as validation set.

The model is fine-tuned on the data that was manually annotated for its pragmatic function (490 utterances) while reinitializing the speaker-like embeddings with four embeddings representing the four pragmatic functions. The model is then trained for an additional 4000 iterations, with checkpoints saved every 500 iterations, to effectively generate the pragmatic functions. An overview of the data used for base training and fine-tuning can be found in Figure 4, while an overview of the training scheme and embeddings used for training can be found in Figure 5. During the process of changing the embedding, there is a temporary loss in speech quality, and the stopping point for training is determined through informal listening tests where

---

[1] https://github.com/NVIDIA/tacotron2

the speech quality sufficiently recovers, but before the model starts overfitting on the data used for fine-tuning. During inference, we synthesize the spontaneous instruction-specific style by assigning the additional weight of $2.5\times$ to the instruction embedding, following [18], to synthesize a distinct pragmatic style.

## 3. Experiments

To investigate the effect of situational and preceding sentence context, we recruited four groups of 23 participants on Prolific who were paid £2.00 and completed the experiments in an average of 10 minutes. Participants were required to be native speakers of English, reside in the United States, and were requested to wear headphones. We conducted four subjective listening evaluations in between-subject design, using four conditions where we vary the type and presence of context which can be summarized as follows:

1. **SituationContext** text description of the situational context of the utterance, question: suitability
2. **PrecedingContext** vocoded ground-truth audio of the preceding utterance from the corpus, question: suitability
3. **FullContext** both types of context, question: suitability
4. **NoContext** TTS samples presented without context, question: naturalness

The audio samples used in the experiment may be found at speech.kth.se/tts-demos/ssw-context.

For the NoContext condition, we conducted a classic naturalness MOS evaluation in which participants were asked: *Listen to the speech sample and rate how natural it sounds to you*, on a scale from *Bad* to *Excellent* presented at the start of the experiment in line with ITU standard P.800.2 [26]. For the stimuli, we synthesized the sentence directly following the excerpts from the held-out recordings. Participants were asked to rate 32 stimuli, 16 of which were synthesized with the spontaneous instruction-specific voice and 16 of which were synthesized with the general spontaneous voice. The stimuli were presented one per screen, without context or transcription, in a random order. We could not use the original recordings as ground truth in this evaluation, as the audio sometimes included overlapping speech from the conversation partner.

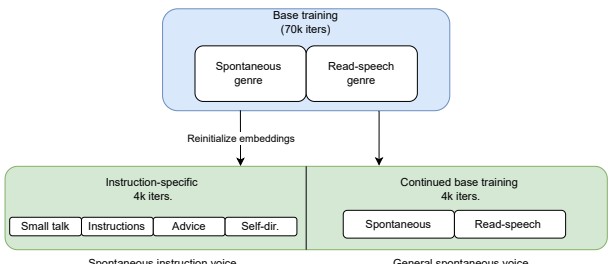

Figure 5: *The training scheme and the embeddings used during training.*

For each of the conditions that included context, we asked participants to rate the suitability. In the SituationContext condition, participants received the following instructions before rating the audio:

1. In this listening experiment, you will hear speech samples from a **game host** who provides **instructions** for a game focused on **designing a room** for a reality TV show similar to Big Brother.

2. The game show host has already interacted with the participants and will now **introduce** the **rules and setup** of the game.

3. The objective of the game is to decorate the room's interior using furniture, and **the players have a budget** of 70,000 crowns to spend in a virtual shop.

Participants were instructed: *Listen to the speech samples and rate how suitable they are for the host of the game show*. We used a five-point rating scale with whole-step increments accompanied with the anchors *5. Very suitable*, *4. Suitable*, *3. Neutral*, *2. Unsuitable* to *1. Very unsuitable*. Participants were then presented with the same stimuli as in the no context situation, with the stimuli from the instruction-specific and spontaneous voices presented on the same screen.

In the PrecedingContext condition, participants were asked to *Listen to the speech samples and rate how suitable they are based on the previous sentence*, where the previous sentence corresponds to the vocoded excerpts from the held-out episodes which occurred directly before the synthesized speech. The same rating scale was used as for the situational context. The stimuli for the spontaneous instruction-specific voice and the general spontaneous voice were presented together, alongside the vocoded excerpt which was labelled *Previous sentence*.

In the FullContext condition, participants were presented with identical instructions to the SituationContext condition, and were asked: *Listen to the speech samples and rate how suitable they are for the host of the game show, also based on the previous sentence*, where the previous sentence and the stimuli were presented identically to the PrecedingContext. The same rating scale was used as for the SituationContext and PrecedingContext.

## 4. Results

In order to assess how the ratings of instruction-specific and spontaneous speech varied in different types of rating tasks we carried out a 2 (system) x 4 (rating task) mixed factorial ANOVA with repeated measures on system. The main effect of system was significant, $F(1,91) = 50.05$, $p < .001$. The spontaneous instruction-specific voice was rated higher than the general spontaneous voice overall. The main effect of rating task was not significant, $F(2,91) = 0.30$, $p = 0.83$. The average ratings given in different rating tasks (collapsed across system) did not significantly differ. However, the interaction between system and rating task was significant, $F(3,91) = 9.53$, $p < .001$, meaning the differences we observed between instruction-specific and spontaneous speech varied in magnitude depending on the rating task. Tests of simple effects of system moderated by rating task show that the difference in ratings of instruction-specific and spontaneous speech were significant in the three suitability rating tasks, $p < .001$, but not for naturalness MOS, $p = 0.13$. Means and standard deviations for each system and task are shown in Table 1.

Tests of simple effects of rating task moderated by system also showed that the mean ratings given in different rating tasks differed for instruction-specific speech, $p < .05$. Post-hoc Tukey tests corrected for multiple comparisons show that participants rated instruction-specific speech significantly higher in the both condition (M=3.84, SD=0.51) and in the situation condition (M=3.85, SD=0.42) than in the no-context condition (M=3.44, SD=0.64), $p < .05$, while none of these means significantly differed from the preceding sentence condition (M=3.79, SD=0.51).

Table 1: *The means and standard deviations for the systems for each suitability condition, with bold indicating a significant difference between the voices.*

| System | MOS | Situation | Preceding | Full |
|--------|-----|-----------|-----------|------|
| Instr. | 3.44 (0.64) | **3.85** (0.42) | **3.79** (0.51) | **3.84** (0.51) |
| Spont. | 3.57 (0.62) | 3.26 (0.67) | 3.07 (0.68) | 3.25 (0.73) |

## 5. Discussion

In this study, we aimed to assess the suitability of an instruction-specific TTS voice and a spontaneous TTS voice for a social robot game guide, by providing participants with context pertaining to the situation or the preceding sentence. Although *appropriateness* is the most commonly used measure in context-based evaluations [9, 10] and evaluations that keep in mind a target application [5, 27], we find that *suitability* has several advantages over *appropriateness*. Although this introduces yet another scale in TTS evaluation, asking participants about *appropriateness* has strong undertones of appropriate content rather than appropriate speech. In our opinion, *suitability* lends itself better to in-context evaluation, as it refers more directly to its relation to the application, whereas *appropriateness* is a more intrinsic attribute of the speech related to its content, rather than its relation to the target application.

These results indicate several aspects regarding the inclusion of context and suitability ratings. The addition of either the situation or the full context impacted the ratings of participants compared to the MOS condition, which was not the case for the PrecedingContext. In general, the spontaneous instruction-specific voice performed significantly better than the general spontaneous voice, except for the MOS condition. No differences were found between the different types of context.

The lack of effect for the PrecedingContext as well as the absence of an additive effect for the situational and preceding context in the FullCondition suggests that participants genuinely evaluated this as context instead of attempting to simply

match the prosody of the TTS with the GT stimulus. It also indicates that participants were not able to infer elements of the situational context from the preceding sentence, which was a possibility especially as some of the excerpts were adjacent to each other in the held-out episode. These results align with the insights and recommendations from [5], suggesting that a stable reference gold standard for speech synthesis cannot exist without considering the situation in which it is embedded.

The results of the post-hoc analysis on the simple effects of rating task moderated by system indicate that introducing preceding sentence context, recorded in a similar situation, did not significantly affect scores compared to a naturalness MOS, unless the situational context was also presented. This further supports the notion that synthesis evaluation should be embedded in realistic applications, and the claim that this embedding increases sensitivity to quality issues. This contrasts with [9], however, who obtained lower scores for the combination of real and synthesized speech compared to synthesized speech in isolation. One possible explanation for this is the close alignment between the context of the original recording scenario to the embedded context of the synthesized speech.

These findings are advantageous for researchers evaluating speech synthesis, as obtaining preceding sentence context is frequently impossible, and labour-intensive to extract even when recordings are available. Embedding the evaluation in a situational context, on the other hand, is an accessible step that appears to assist participants in envisioning the scenario in which they would interact with the evaluated TTS voice(s).

## 6. Conclusion

In this paper, we evaluated the suitability of an instruction-specific TTS voice and a spontaneous TTS voice both trained on conversational data, for the role of a social robot game guide. The evaluation utilized subjective listening experiments that included situational and preceding sentence context. In the three conditions with context, the spontaneous instruction-specific voice received significantly higher ratings than the general spontaneous voice, whereas there was no significant difference in the no context condition. Additionally, there was no discernible difference between the various types of context. These results contribute to a growing body of work on the importance of embedding speech synthesis evaluations within the situational setting of their application. We can conclude that this type of situational embedding is equally beneficial as including ground truth audio excerpts as context. Although interactional designs may be the ultimate use-case scenario, simply describing a situation which people can envision can already assess more nuanced differences between TTS versions, compared to a classic naturalness MOS evaluation.

Future work could further investigate ways in which evaluations can be socially embedded even when using crowdsourcing platform-based listening tests, especially examining the differences with interaction design-based evaluations. An additional opportunity for research lies in creating methods that aid speech synthesis models in selecting the correct pragmatic function for a given situation.

## 7. Acknowledgments

This research was supported by the Swedish Research Council projects Connected (VR-2019-05003), Perception of speaker stance (VR-2020-02396), and the Riksbankens Jubileumsfond project CAPTivating (P20-0298).

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
