# OpenReview forum: "Situating Speech Synthesis: Investigating Contextual Factors in the Evaluation of Conversational TTS"
_Interspeech.org/2023/Workshop/SSW — SSW12_

### Official Review · Reviewer_mYYu · 2023-06-05
**Situating Speech Synthesis**

**Rating:** 7
**Confidence:** 5

**Review:**

This paper investigates the subjective evaluation of TTS using a context. The context can be either information about the situation or simply the preceding sentence from the corpus.
The TTS system used is derived from a Tacotron2 model including two kinds of embedding (one for the pragmatic function and one for the type of speech - spontaneous vs. read). This part could be explained better. In particular, the way the embeddings are used in the system is not always clear.
4 subjective experiments are conducted to evaluate the influence of situation context, sentence context, the use of both, and the absence of context. Interestingly, authors propose to evaluate the "suitability" and not the "appropriateness" which does make sense. I think that this kind of evaluation could bring interesting insights in the way TTS is perceived by users. It is clear complementary to existing evaluations.

Overall, the paper is well written and technically solid. The idea is simple and seems to work quite well. I recommend to accept this paper.


Typos:
- you would need given
- which can especially important
- more nouanced

---

### Official Review · Reviewer_Pi54 · 2023-06-05
**SSW23 review for submission 29**

**Rating:** 7
**Confidence:** 3

**Review:**

The paper presents an experiment to assess the merit of including contextual factors in the evaluation of conversational TTS. The paper tries whether (a) a textual description of the context or (b) a preceding vocoded natural utterance as context or (c) both are useful to evaluate the suitability of TTS voices in addition to standard MOS. The paper evaluates two different TTS systems, one trained on instructions (A) and another on spontaneous speech (B). It turns out that both voices are not significantly different in terms of MOS but all context including evaluations show a significant advantage of system A (which, surprisingly, is not the "spontaneous" but the more standard "instructional" system).
This can be interpreted as the context yielding information to the user of the system that makes them value system A as better. However, it's somewhat unclear, why system A would be able to pick up this context and better render the speech.
Under the assumption that the "contextual" evaluation is better suited for evaluating conversational TTS (and the paper cites ample evidence for this), the paper finds that there are no significant differences between the ways that context was included (a, b or c). Therefore, the paper argues, it is not necessary to include vocoded preceding utterances as context but a mere textual description of the context is sufficient.
Pros:
- concise experiment that yields evidence that a "simple" way of extending MOS for context information is as good as more involved setups.
Cons:
- it is surprising (to say the least) that the spontaneous TTS is inferior to the instructional TTS in the conversational setting. This may be linked to some issues in the TTS training setups which somewhat calls the overall results into question.
- It is unclear how either system would be able to include context. The paper should discuss how different contexts for the same TTS rendering yields different results (or not) and should give a good reason if such experiments were not included.

---

### Decision · Program_Chairs · 2023-06-14

**Decision:**

Accept

**Comment:**

SSW2003 received 45 papers. The acceptance rate is 82%. We are pleased to inform you that your paper has been accepted by the SSW2023 Program Committee. Please read the reviews carefully and submit your camera-ready paper by June 28th. Most reviewers performed a detailed review. Please answer to their questions and consider their comments. Note that camera-ready papers are credited with one extra page to allow authors to consider reviewers’ suggestions. So max 7 pages in total including figures & refs.
The deadline for submitting the revised version (with full non-anonymized authors and refs!) is 28th June.